# Development of an Ex Vivo Porcine Eye Model for Exploring the Pathogenicity of *Acanthamoeba*

**DOI:** 10.3390/microorganisms12061161

**Published:** 2024-06-06

**Authors:** Ming-Der Shi, Ko-Chiang Sung, Jian-Ming Huang, Chun-Hsien Chen, Yu-Jen Wang

**Affiliations:** 1Department of Clinical Laboratory, Chest Hospital, Ministry of Health and Welfare, Tainan 717, Taiwan; tn20132@ccd.mohw.gov.tw (M.-D.S.); tn50282@ccd.mohw.gov.tw (K.-C.S.); 2School of Medicine, National Tsing Hua University, Hsinchu 300, Taiwan; jmhuang@life.nthu.edu.tw; 3Institute of Molecular and Cellular Biology, National Tsing Hua University, Hsinchu 300, Taiwan; 4Institute of Basic Medical Sciences, College of Medicine, National Cheng Kung University, Tainan 701, Taiwan; greatwall91983@gmail.com; 5Department of Parasitology, School of Medicine, China Medical University, Taichung 404, Taiwan

**Keywords:** ex vivo model, *Acanthamoeba*, pathogenicity, keratitis

## Abstract

*Acanthamoeba*, a widely distributed free-living amoeba found in various environments, is an opportunistic pathogen responsible for causing *Acanthamoeba* keratitis, a condition that may lead to blindness. However, identifying the pathogenicity of *Acanthamoeba* is challenging due to its complex life cycle, ability to adapt to different environments, variable virulence factors, and intricate interactions with the host immune system. Additionally, the development of an effective model for studying *Acanthamoeba* pathogenicity is limited, hindering a comprehensive understanding of the mechanisms underlying its virulence and host interactions. The aim of this study was to develop an ex vivo model for *Acanthamoeba* infection using porcine eyeballs and to evaluate the pathogenicity of the *Acanthamoeba* isolates. Based on slit lamp and biopsy analysis, the developed ex vivo model is capable of successfully infecting *Acanthamoeba* within 3 days. Histopathological staining revealed that clinical isolates of *Acanthamoeba* exhibited greater corneal stroma destruction and invasion in this model than environmental isolates. Our results highlight the importance of an ex vivo porcine eye model in elucidating the pathogenesis of *Acanthamoeba* infection and its potential implications for understanding and managing *Acanthamoeba*-related ocular diseases.

## 1. Introduction

*Acanthamoeba*, a genus of ubiquitous free-living amoebae, is globally distributed across diverse natural habitats, including soil, fresh water, and marine water [1,2]. Clinical isolates of *Acanthamoeba* serve as etiological agents of *Acanthamoeba* keratitis (AK), an ocular disease characterized by pain and progressive symptoms that can lead to vision loss [3]. AK patients commonly exhibit symptoms such as photophobia, stromal infiltration with a ring-like appearance, epithelial defects, and eyelid edema, accompanied by considerable pain [4]. Although there is a strong association between AK and contact lens wear, AK can occur in noncontact-lens wearers, suggesting potential variations in pathogenicity among different *Acanthamoeba* isolates [5]. In some individuals with AK, *Acanthamoeba* can invade the corneal stromal layer and persist within it in the form of cysts, ultimately leading to blindness. The invasion of the corneal stroma can be attributed to the degradative effects induced by *Acanthamoeba*-released proteases and the ability of *Acanthamoeba* to migrate within the tissue [6,7]. However, previous in vitro experiments have revealed substantial variations in pathogenicity among different isolates of *Acanthamoeba* [8], emphasizing the importance of discerning the pathogenicity of *Acanthamoeba* for clinical diagnosis and treatment.

Studies on the use of in vivo models to investigate the development of AK caused by *Acanthamoeba* have been published since 1972 [9]. However, the optimal animal models and treatment approaches have yet to be established. This indicates the presence of various unstable factors in the AK models. The most commonly used method for AK drug treatment research is intracorneal injection [10,11]. This approach simulates the invasion of *Acanthamoeba* into the corneal stromal layer; however, it is not entirely representative, and most protozoa are likely to be lost during the injection process. To simulate real infection scenarios, miniature contact lenses have also been developed and implanted in Chinese hamsters [12]. However, this is not an ideal application for an infection model, and the efficiency of infection may depend on the immune response of the experimental animals.

In this study, we developed an effective and accessible ex vivo model using porcine eyeballs that closely resemble the human corneal structure. Slit lamp examination was used to assess the extent of ocular damage and surface disruption. Histopathological staining was used to evaluate the depth of *Acanthamoeba* invasion into the corneal stromal layer and the pathological features of protease-dissolved tissue. Finally, we compared the pathogenicity between environmental and clinical isolates of *Acanthamoeba* in this model. This model can be widely used in future research related to the treatment of AK and the investigation of *Acanthamoeba* pathogenicity.

## 2. Materials and Methods

### 2.1. Culture of Acanthamoeba Protozoa

The *Acanthamoeba castellanii* strains ATCC-30010 and ATCC-50492 were obtained from ATCC (Manassas, VA, USA). Clinical isolates were collected from patients diagnosed with *Acanthamoeba* keratitis, as previously described. To culture the *Acanthamoeba* strains, protease peptone-yeast extract-glucose (PYG) medium was used. PYG medium was prepared by combining 20 g of peptone, 2 g of yeast extract, 18 g of glucose with 1 g of sodium citrate dehydrate, 0.98 g of MgSO_4_ × 7H_2_O, 0.355 g of Na_2_HPO_4_ × 7H_2_O, 0.34 g of KH_2_PO_4_, and 0.02 g of Fe[NH_4_]_2_[SO_4_]_2_ × 6H_2_O in a total volume of 1000 mL of distilled water (pH 6.5). The cultures were maintained at 28 °C in cell culture flasks and subsequently washed with Page’s modified Neff’s amoeba saline (composed of 1.2 g NaCl, 0.04 g MgSO_4_ × 7H_2_O, 0.03 g CaCl_2_, 1.42 g Na_2_HPO_4_, and 1.36 g KH_2_PO_4_ per 1 L ddH_2_O).

### 2.2. Porcine Eye Pretreatment

The porcine eye was selected as an ex vivo animal model to simulate *Acanthamoeba* keratitis in humans. With numerous anatomical similarities to human eyes [13,14], porcine eyes have been utilized as a model for studying wound healing, as demonstrated by Nileyma Castro et al. [15]. Porcine eyeballs were obtained from a nearby slaughterhouse. To maintain tissue moisture and integrity, all animals were euthanized on the same day as the study, and the eyeballs were stored in 0.9% NaCl isotonic solution at room temperature. The time between pig slaughter and testing was kept below 12 h.

Upon arrival at the laboratory, all porcine eyeballs had their optic nerve, muscle tissue, and conjunctiva removed using ophthalmic surgical instruments. Subsequently, the eyeballs were immersed in 5% povidone iodine solution for 2 min to disinfect them. Finally, they were rinsed with sterile physiological saline solution until no visible residue of povidone iodine remained.

### 2.3. Acanthamoeba Infection

To simulate *Acanthamoeba* keratitis in humans, porcine eyeballs with an intact corneal surface were used as a negative control. The positive control consisted of porcine eyeballs in which 3 × 10^5^ *Acanthamoeba* were injected into the cornea using a 30G needle. The control group had sterile and nonprescription contact lenses covering the corneal surface. In the experimental groups, the corneal surface was abraded with 100-grit sandpaper or scraped with a 30G needle. After corneal integrity disruption was induced, nonprescription contact lenses seeded with 3 × 10^5^ *Acanthamoeba* were placed on the corneal surface. Finally, all porcine model eyes were placed cornea-side up in sterile containers lined with moistened sterile gauze soaked in physiological saline solution. Throughout the infection process, they were kept in a temperature-controlled incubator at 28 °C. Three days after infection, all contact lenses on the corneas of the porcine eyes were removed, and the corneas were surgically excised for subsequent evaluation. All experiments were repeated at least three times.

### 2.4. Slit Lamp Examination

The excised corneas were examined using a slit lamp (SL-1600; Nidek Co., Ltd., Aithi, Japan) at the southern office of the National Eye Bank of Taiwan in National Cheng Kung University Hospital. The assessment parameters included the integrity of the corneal surface and its transparency.

### 2.5. Histological Evaluation

Following the experimental trial or assignment as a control, each eyeball underwent immediate gross examination to identify any visible signs of injury. Subsequently, the eyeballs were fixed in a 10% neutral buffered formalin solution (Sigma-Aldrich, St. Louis, MO, USA). Within a timeframe of 12 h, all specimens were grossly examined, paraffin-embedded, sectioned at a thickness of 5 microns, and stained with hematoxylin-eosin (H&E).

### 2.6. Comparison of Acanthamoeba Strain Infections

To observe and compare the differences in infection among different *Acanthamoeba* isolates, we selected a 30G needle to induce corneal surface disruption in the ex vivo porcine model eyes, followed by infection using the method mentioned above. The assessment included observing the pathological changes caused by different *Acanthamoeba* isolates within the corneal stromal layer using the aforementioned tissue staining method. Additionally, we captured and measured the extent of *Acanthamoeba* dispersion using an Olympus microscope (Tokyo, Japan). The extent of dispersion was measured and calculated using Olympus microscope software (driver v3.6 & ImagePro Plus software) by determining the distance between at least 10 observed *Acanthamoeba* cells in the stromal layer and the corneal epithelial cell layer.

### 2.7. Statistical Analysis

The data are presented as the mean ± standard deviation (SD). Statistical analysis was performed using unpaired two-tailed Student’s *t* tests for all comparisons. A significance level of *p* < 0.05 was considered statistically significant. GraphPad Prism version 5.0 software (La Jolla, CA, USA) was used for data calculation and analysis.

## 3. Results

### 3.1. Processing of Porcine Eyeballs as an Ex Vivo Model for Acanthamoeba Infection

All porcine eyeballs were obtained from a nearby slaughterhouse and immediately removed after pig slaughter. Upon arrival at the laboratory, the acquired porcine eyeballs were promptly dissected to remove any surrounding tissues (Figure 1A). Pretreated porcine eyeballs were also immediately subjected to experimental procedures, and to prevent desiccation during the experiments, sterile moistened gauze was placed at the bottom of the containers. This moistened gauze served the dual purpose of maintaining hydration and securing the porcine eyeballs in an upright position with the cornea facing upward (Figure 1B). Furthermore, regardless of whether *Acanthamoeba* infection occurred, nonprescription contact lenses were placed on the corneal surface to simulate the most common route of AK, which occurs through contact lens wear [16]. After completing the pre-treatment of porcine eyeballs, this ex vivo model was placed in an incubator at 28 °C for the experiment. Although the surface temperature of the human eye is higher than the experimental temperature, the model was designed to facilitate a mild infection and evaluate the pathogenicity of different isolates. To prevent evaporation and subsequent dehydration of the porcine eyeballs, and considering that higher temperatures might affect the pathological invasion of *Acanthamoeba* [17], we used the same temperature as for in vitro culture to set the infection conditions.

### 3.2. Needle Scraping as an Effective Method to Simulate Corneal Injury and Infection in Porcine Models

Corneal surface injury is a common risk factor for corneal infection [18]; therefore, to simulate corneal surface injury, we employed two different methods: abrasion with 100-grit sandpaper and scraping with a 30G needle. Following corneal injury, nonprescription contact lenses seeded with 3 × 10^5^ *Acanthamoeba* ATCC-50492 were placed on the cornea for infection. After 3 days of infection, it was evident that corneas scraped with a 30G needle exhibited visible corneal haze. This pathological appearance was similar to that of the positive control group, in which *Acanthamoeba* infection was directly injected using a needle. However, porcine eyeballs subjected to corneal abrasion with 100-grit sandpaper did not display this characteristic feature (Figure 2). Next, we evaluated the appearance of the corneal infection using a slit lamp. Following the experimental trial, the corneal scleral rim was removed using scleral scissors, and any iris tissue on the back of the cornea was removed using a spatula. The corneal scleral rim was evaluated using a slit lamp, and the results demonstrated significant corneal opacity in both the positive control group and the group with corneal scraping using a needle. However, only the corneas subjected to needle scraping displayed noticeable corneal surface irregularities (Figure 3). These results indicate that significant pathological changes can be observed within a short timeframe of 3 days when the corneal surface is scraped using a needle in this model.

### 3.3. Histological Confirmation of Acanthamoeba Invasion in Needle-Scraped Corneas

However, successful infection and invasion of AK lesions still need to be confirmed through histological evaluation of tissue sections. Therefore, immediately after examination with a slit lamp, the corneal scleral rim was promptly processed for embedding, sectioning, and staining with hematoxylin-eosin. The results of tissue section staining were consistent with the changes observed in the corneal appearance and slit lamp examination. Successful *Acanthamoeba* invasion was observed exclusively in corneas subjected to needle scraping. In the group with needle-scraped corneas, we clearly observed parasites adhering to the corneal epithelial cell layer without fully invading the cornea. Additionally, some parasites were observed to have invaded the stromal layer, leading to tissue dissolution within the stromal tissue. Although tissue dissolution has been observed in previous studies using corneal injection infection methods, this approach is less similar to real infection scenarios, and all parasites aggregate within a small, localized area within a short timeframe (Figure 4).

### 3.4. Evaluation of the Pathogenicity of Acanthamoeba Isolates Using a Needle-Scraped Porcine Eye Model

After confirming that corneal scraping with a needle could simulate AK occurrence within a short timeframe, we infected porcine eyeballs with the *Acanthamoeba* environmental isolate ATCC-30010 and clinical isolates ATCC-50492, NCKUH-B, and NCKUH-D using the same method. According to American Type Culture Collection (ATCC) documentation, ATCC-50492 was isolated from a patient with AK in India, while NCKUH-B and NCKUH-D were isolated from AK patients previously infected during contact lens wear in our study [19]. With the needle-scraped porcine eye model, we conducted preliminary evaluations to assess whether different isolates from various sources exhibit variations in pathogenesis. Based on the results of histological tissue staining, we observed significant tissue dissolution in the corneal stromal layer for the clinical isolates ATCC-50492, NCKUH-B, and NCKUH-D. Additionally, there was a noticeable zone of tissue dissolution surrounding the *Acanthamoeba* cells. However, although the environmental isolate ATCC-30010 also successfully invaded the corneal stromal layer, the resulting zone of tissue dissolution was not as prominent (Figure 5). The depth of invasion of *Acanthamoeba* cells into the corneal stromal layer during the course of human AK can impact the effectiveness of topical medications for treating AK. Therefore, in the following steps, we utilized software to measure, under microscopic observation, the distance between the location of *Acanthamoeba* cells in tissue sections and the corneal epithelial cell layer. After comparing the depth of invasion, the average invasion depths of the clinical isolates were measured as follows: ATCC-50492 = 124.36 µm, NCKUH-B = 145.32 µm, and NCKUH-D = 305.27 µm, which were significantly greater than the invasion depth of the environmental isolate ATCC-30010, which measured 90.34 µm (Figure 6). Additionally, in line with our previous findings of greater cytotoxicity of NCKUH-D in in vitro experiments [20], NCKUH-D also increased the extent of corneal tissue invasion. The data demonstrate that this ex vivo porcine eye model, which utilizes corneal surface scraping with a needle followed by *Acanthamoeba* infection through contact lens wear, is effective for studying *Acanthamoeba* pathogenicity and drug therapy related to AK.

## 4. Discussion

The development of an effective animal model of *Acanthamoeba* keratitis (AK) is crucial for conducting in-depth investigations into the molecular biology, pathology, and immunology of this disease. Additionally, such a model plays a vital role in evaluating and testing new pharmacologic agents designed to control AK [21,22]. Due to its comparative rarity in contrast to infections caused by other pathogens, it is crucial to employ the appropriate model and the most effective methodology when studying corneal *Acanthamoeba* infection. Previous studies have utilized pigs, rabbits, and Chinese hamsters to induce AK to investigate AK [23,24,25]. However, when comparing the AK infection models established using rats and mice, it was found that while mice were more susceptible to AK induction than rats, they also exhibited higher mortality rates and increased technical difficulty during the procedure [11]. This is likely due to variations in immune response levels among different species, which can result in the instability of the induced model. The instability of the model resulting from variations in immune response levels can impact its subsequent applications in therapeutic and pathogenicity studies.

The human cornea also has several physical and chemical mechanisms that enable it to resist infection by external pathogens [26]. The amoebae would normally be removed through the blinking action of the eyes [27]. Therefore, in previous AK animal infection models, to prevent this mechanism from causing failure in establishing the infection model, the eyelids of the infected animals were surgically sutured after administering the infection [28]. However, this additional suturing to prevent model induction failure due to blinking likely induces stress and physiological pressure on the infected animals, which could further alter their immune response and affect the infection rate of the inoculated *Acanthamoeba*. In addition, tears contain various antimicrobial components, including proteins and lipids. Previous studies have shown that mammalian tears contain an average of about 11.06 μg/μL of protein [29]. Human tears even contain almost 2000 tear-specific proteins [30] and several hundred types of lipids [31]. However, these factors can lead to inaccuracies when studying AK-related issues in models. For instance, in our previous study, we demonstrated that mucin in tears can counteract the efficacy of chlorhexidine digluconate (CHG), a first-line treatment for AK [32]. Such interfering factors can potentially introduce errors in evaluating drug toxicity in animal models for drug treatment. In this research paper, we also cleaned the surface of porcine corneas by disinfection and rinsing. This is because in our previous studies, various microorganisms were present on the corneal surface [33], and we have also confirmed that the presence or absence of normal ocular flora may direct the infected *Acanthamoeba* toward phagocytizing bacteria rather than causing cytotoxicity to the cell layer [34]. Therefore, narrowing down potential interfering factors in the infection model will contribute to studying the AK issues we aim to explore, ensuring that the research focus remains clear and unambiguous.

In this study, we attempted to establish an ex vivo model that eliminates the interference of varying immune response levels among species. In the absence of an immune response, *Acanthamoeba* cells can steadily invade the corneal stromal layer through the corneal epithelial cell layer. This infection model mimics the course of AK infection in humans and eliminates the need to sacrifice a large number of animals with successful infections due to individual variations in immune responses. By utilizing the corneal epithelial cell layer as the starting point of infection, *Acanthamoeba* cells can spread within the corneal stromal layer, mimicking the real infection scenario, rather than aggregating in large quantities, as observed in the positive control group of this study. The diffusion and invasion of pathogens within the corneal stromal layer are crucial for studying drug treatment and the underlying pathological mechanisms involved. The extensive aggregation of *Acanthamoeba* cells within the stromal layer without disruption of the epithelial cell layer may impact the efficacy of chemotherapeutic agents being investigated. Furthermore, in terms of studying pathological mechanisms, the tissue dissolution caused by *Acanthamoeba* may be limited to specific regions rather than affecting the entire corneal tissue. The ex vivo porcine eye model established in this study effectively eliminates these confounding factors and simulates the real-life scenario of AK infection.

Pigs have been widely utilized as experimental models in biomedical research due to their anatomical and physiological similarities to humans. Numerous studies have explored various aspects of pig eyes, including parameters related to the entire eyeball [35,36], retina [37,38], cornea [39], limbus [40], and the lacrimal gland [41]. Furthermore, the porcine conjunctiva exhibits similarities to that of humans. As in humans, porcine conjunctival goblet cells are distributed across the entire conjunctival epithelium [42]. The porcine cornea, which is affected by AK lesions, consists of an epithelium, stroma, Descemet’s membrane, and endothelium. However, the existence of Bowman’s layer remains controversial, as does its structural resemblance to that of humans, especially in terms of the epithelial cell layer [39]. The collagen fibril structure and arrangement within the porcine corneal stroma are more similar to those of humans, which contributes to the greater stability of preserving human and porcine corneas compared to that of preserving rabbit corneas [43]. The anatomical differences between the selected species will also impact the invasion rate of the pathogen from the external to the internal ocular tissues and the evaluation of drug permeation rates in eye drops.

Finally, based on the experimental results evaluated in this study, we used a needle-scraped porcine eye model to assess the pathogenicity of *Acanthamoeba* isolates from various sources, including soil and AK patients’ cornea. By excluding relevant complex factors, including physiological and mechanical pathogen-resistant mechanisms, we focused solely on tissue structural damage. The results showed that clinical isolates exhibited greater tissue dissemination compared to environmental isolates. This finding is consistent with previous studies that reported stronger corneal tissue damage by clinical isolates [8]. However, for future *Acanthamoeba*-associated research and pathogenicity assessment, the porcine eye model developed in this study is easier to obtain and presents fewer ethical concerns. The exact reasons for the differences in pathogenicity among *Acanthamoeba* isolates from different sources remain unclear. Previous studies have shown significant differences in the immunoreactivity profiles of proteins when analyzing genotypes T4, T5, T6, T7, T9, T11, and T12 [44]. Additionally, proteomic analysis has revealed that even within genotype T4, there are differences in the secreted proteins between environmental and clinical isolates [45]. These factors may contribute to the varying degrees of tissue dissolution during opportunistic infections, resulting in differences in corneal spreading distances. Therefore, the ex vivo model developed in this study provides an effective platform for evaluating *Acanthamoeba* strains of different genotypes or sources, facilitating the identification of key factors responsible for pathogenicity differences.

## 5. Conclusions

In conclusion, our study provides a rapid and effective ex vivo model for AK. This model eliminates the need for sacrificing experimental animals and offers convenience and cost-effectiveness. The ex vivo infection model also effectively avoids immune interference of varying potencies. By studying human AK infections and the presence of similar anatomical structures, this model realistically simulates clinical pathological characteristics. By infecting contact lenses seeded with *Acanthamoeba* cells, we can replicate the actual progression of AK from the epithelial cell layer to the stromal layer. The results of our study can be widely utilized in future research on drug treatments and understanding the pathological mechanisms of AK.

## Figures and Tables

**Figure 1 microorganisms-12-01161-f001:**
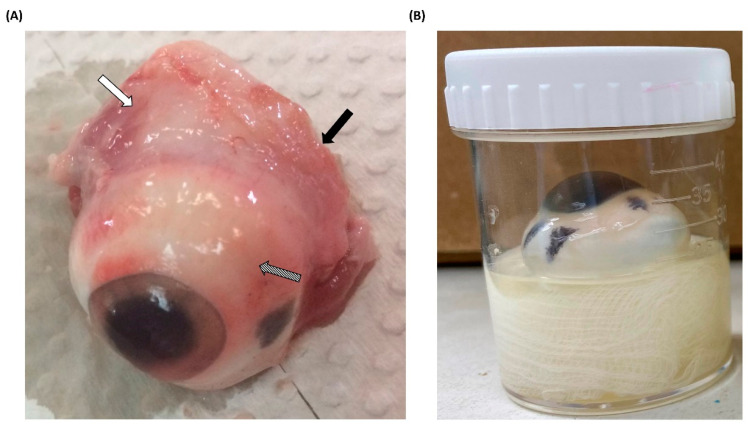
Pretreatment and infection methods for porcine eyeballs (**A**) Porcine eyeballs obtained from freshly slaughtered pigs were immediately transported to the laboratory. The surrounding muscle tissue (white arrow), optic nerve (black arrow), and conjunctiva (diagonal arrow) were removed using surgical instruments. (**B**) Pretreated and disinfected porcine eyeballs were placed in sterile containers with moistened gauze, ensuring that the cornea faced upward. Nonprescription contact lenses were then applied to the corneal surface.

**Figure 2 microorganisms-12-01161-f002:**
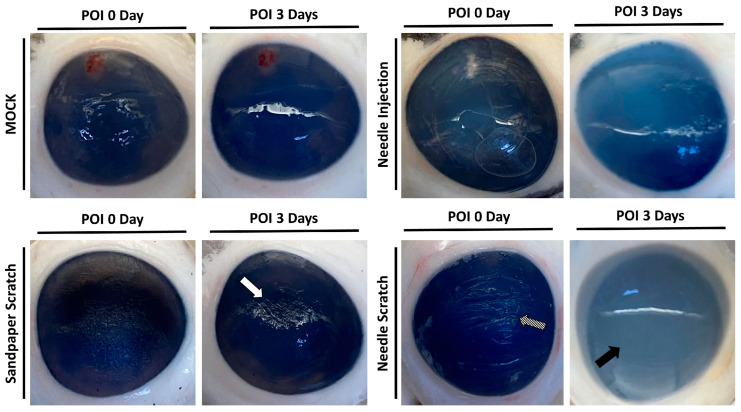
Appearance of porcine cornea after infection. Following the inoculation of *Acanthamoeba* cells (ATCC-50492) at a concentration of 3 × 10^5^, porcine eyeballs were retrieved from sterile containers after 3 days. The cornea, abraded with 100-grit sandpaper, displayed visible abrasion marks but no corneal opacification (white arrow). On the other hand, the cornea subjected to scraping with a 30G needle exhibited clear signs of abrasion (diagonal arrow) and significant corneal opacification (black arrow). The MOCK group represented the negative control without infection, while the needle injection group represented the positive control, with an equivalent dose of the infectious pathogens injected into the corneal stromal layer. All of the experimental tests were replicated three times.

**Figure 3 microorganisms-12-01161-f003:**
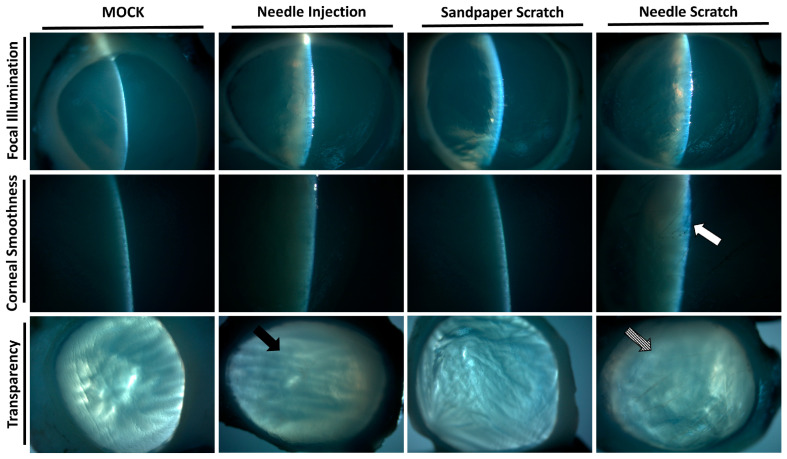
Slit-lamp examination of the infected corneal scleral rim. After 3 days of infection in porcine eyeballs, the cornea was excised using surgical instruments, and slit-lamp examination was performed to evaluate the corneal surface and transparency. The cornea abraded with 100-grit sandpaper showed an intact surface and suitable transparency. In contrast, the cornea scraped with a 30-gauge needle displayed a damaged surface (white arrow) and uniform opacity (diagonal arrow). The surface of the cornea in the needle-injected group remained intact, but localized opacity was observed (black arrow). All of the experimental tests were replicated three times.

**Figure 4 microorganisms-12-01161-f004:**
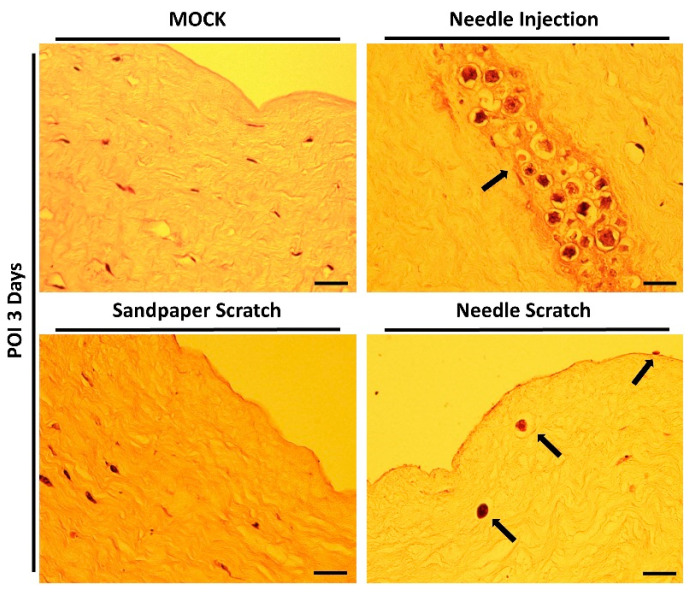
Histological staining of the infected cornea. After 3 days of infection, the corneal scleral rim was excised and subjected to hematoxylin-eosin staining for histopathological analysis. In corneas abraded with 100-grit sandpaper, no invasive *Acanthamoeba* cells were observed within the stromal layer. In contrast, corneas scraped with a 30G needle showed active disruption of the epithelial cell layer and invasion of the stromal layer by *Acanthamoeba* cells (black arrows). In the needle injection group (positive control), *Acanthamoeba* cells were localized and distributed at the site of needle injection (black arrow). Scale bar = 20 μm.

**Figure 5 microorganisms-12-01161-f005:**
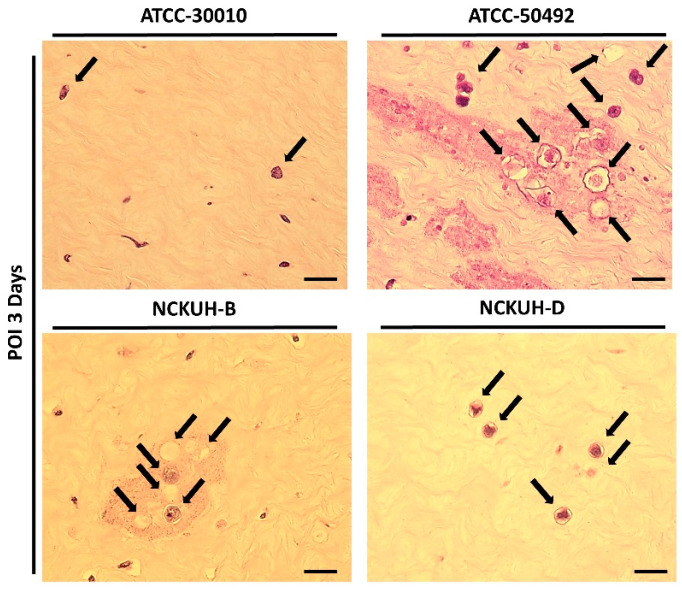
Comparison of the pathological features of different *Acanthamoeba* isolates in the ex vivo porcine eye model with 30G needle scraping. All *Acanthamoeba* isolates (black arrows) exhibited invading cells in the corneal stromal layer. Among the isolates, the clinical isolates (ATCC-50492, NCKUH-B, and NCKUH-D) had more invading cells in the stromal layer than did the environmental isolate (ATCC-30010). Additionally, significant tissue dissolution was observed in the corneal histopathological staining of the clinical isolates. Scale bar = 20 μm.

**Figure 6 microorganisms-12-01161-f006:**
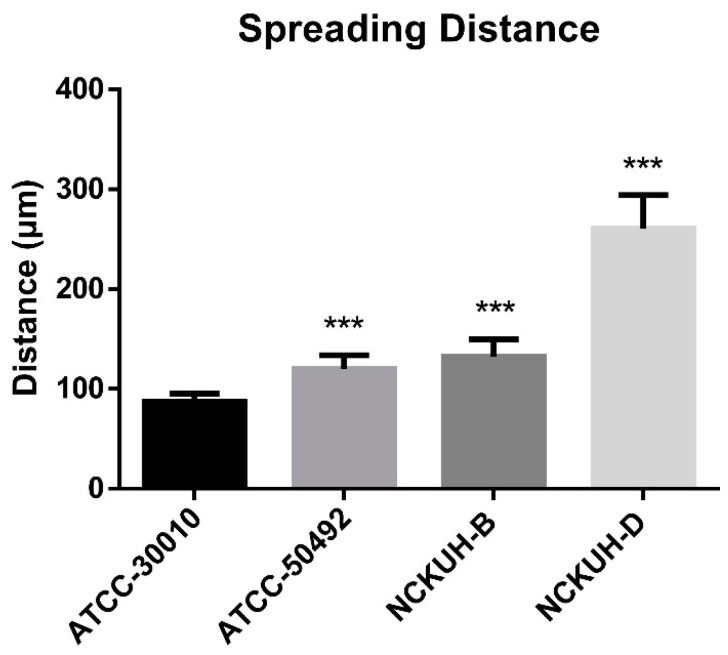
Comparison of corneal invasion depth among different *Acanthamoeba* isolates. Under microscopic observation and imaging software, the average distance of 10 *Acanthamoeba* cells from the corneal epithelial cell layer was calculated. Compared with the environmental isolates, the clinical isolates exhibited significantly greater invasion depth. Among the clinical isolates, the NCKUH-D strain, which displayed greater cytotoxicity, demonstrated the greatest invasion depth. Statistical significance was determined by a Student’s *t* test. *** *p* < 0.001.

## Data Availability

The original contributions presented in the study are included in the article, further inquiries can be directed to the corresponding author.

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
