# Peer review of "Development of an Ex Vivo Porcine Eye Model for Exploring the Pathogenicity of *Acanthamoeba"

_microorganisms, 2024, doi:10.3390/microorganisms12061161_

Round 1
Reviewer 1 Report
Comments and Suggestions for Authors
This paper describes the establishment of an ex vivo model for Acanthamoeba Keratitis using Pig eyeball. The paper is very well written and clear. However, I have a few questions.
Why were the eyeballs and amoebae incubated at 280C when the surface of the human eye is 34-35oC? I presume that this temperature was chosen to limit evaporation and subsequent dehydration of the eyeball?
The eyeballs are sterilised with a povidone iodine solution and then thoroughly rinsed in a 0.9% saline solution. This process will remove the tears. Mammalian tears contain about 11.06 μg/μl of protein (Dor et al, 2019). Many of these have antimicrobial properties. There are almost 2000 tear specific proteins in humans (Zhou et al, 2012) and a few hundred individual lipid types have been isolated from tear fluids (Butovich, 2013). The lack of trears and any influence they may have will not be apparent in the assay as it is unless of course they can be added back to the system if they are available from another source.
The temperature difference and the lack of eye fluids (tears) make this Pig Eye model a limited but useful model, but these two deficits are likely to be important. These deficits should be mentioned and discussed by the authors in the paper.
It is important to note that one of the Acanthamoeba strains used, ATCC 30010 is widely regarded as being non-pathogenic (a fact known to this group, Huang et al 2016) and so this should be highlighted as being a negative control? The fact that this strain showed an ability to penetrate the cornea may complicate the interpretation a bit? The Legend to figure 6 did not mention if the cornea had been abraded.
Line 4. It’s better to change this to “Acanthamoeba keratitis, a condition that may lead to blindness.” As fortunately not all AK cases result in blindness.
References
Butovich, I.A. (2013) Tear film lipids. Exp Eye Res.117, 4–27.
Dor, M., Eperon, S., Lalive, P. H., Guex-Crosier, Y., Hamedani, M., Salvisberg, C., & Turck, N. (2019). Investigation of the global protein content from healthy human tears. Exp Eye Res, 179, 64-74.
Huang, J. M., Lin, W. C., Li, S. C., Shih, M. H., Chan, W. C., Shin, J. W., & Huang, F. C. (2016). Comparative proteomic analysis of extracellular secreted proteins expressed by two pathogenic Acanthamoeba castellanii clinical isolates and a non-pathogenic ATCC strain. Experimental parasitology, 166, 60-67.
Zhou L, Zhao SZ, Koh SK, et al. (2012). In-depth analysis of the human tear proteome. J Proteomics. 75(13), 3877–3885.
Reviewer 2 Report
Comments and Suggestions for Authors
Sung et al have developed an ex vivo model for AC keratitis using porcine eye balls. the paper is descriptive, showing that AC will invade the cornea stroma after abrasion of the eyeball surface. I would suggest some minor modifications to the paper.
1. Abstract - line 23-24, needs re-phrasing. the model is not infecting AC, it's the other way around.
2. Introduction and methods both fine.
3. Results and Discussion - as this is a descriptive study, I would recommend that you merge the results and discussion into one section. The paper is not data rich and a combined Results/Discussion section would make more sense.
4. Lines 134-145 can be deleted, as they repeat the methods and legend to fig. 1
5. Why was the dose of 3x10^5 AC chosen to infect the eyes? Please provide rationale.
6. Figures 2 and 3 are difficult to follow, without labelling the structures/effects with arrows and accompanying explanations. Please mark up.
7. The authors should offer some discussion of the pathogenicity of AC when they compare the clinical and environmental isolates. What do they think could account for the differences?
8. More discussion is needed about the possibilities that the model offers. Can you sue it to look at innate immune responses? Are macrophages present do you know? What other biological measurements could you make, e.g. cell apoptosis, etc. Also, what are the limitations of the model, compared to animals that can mount immune responses, etc.
Comments on the Quality of English LanguageEnglish is very good, no issues, except the one sentence in abstract that needs editing.
